# Not a Dead-End Host: First Confirmed Persistent Microfilaremia in Human *Dirofilaria repens* Infection

**DOI:** 10.3390/microorganisms13102263

**Published:** 2025-09-26

**Authors:** Martina Perešin Vranjković, Anamarija Vitko Havliček, Martina Kramar, Mirjana Balen Topić, David Beck, Daria Jurković Žilić, Ema Gagović, Relja Beck

**Affiliations:** 1University Hospital for Infectious Diseases “Dr. Fran Mihaljević”, 10000 Zagreb, Croatia; mperesin@bfm.hr (M.P.V.); anmari2211@gmail.com (A.V.H.); martina.kramar1@gmail.com (M.K.); mbalen@bfm.hr (M.B.T.); 2School of Medicine, University of Zagreb, 10000 Zagreb, Croatia; dabeck025@gmail.com; 3Department for Bacteriology and Parasitology, Croatian Veterinary Institute, 10000 Zagreb, Croatia; jurkovic@veinst.hr (D.J.Ž.); gagovic@veinst.hr (E.G.)

**Keywords:** *Dirofilaria repens*, microfilaremia, human, diagnosis, Croatia

## Abstract

We report the first confirmed case of persistent microfilaremia in a human host infected with *Dirofilaria repens*. A 54-year-old woman from an endemic area in Croatia presented with peripheral eosinophilia and dermatological symptoms. Over four months, microfilariae were repeatedly detected in her blood using thick smears and Knott’s test, and the diagnosis was molecularly confirmed via *COI* gene sequencing and detection of *Wolbachia* endosymbionts. This case provides compelling evidence that *D. repens* can sustain a complete or near-complete life cycle in humans under specific conditions. Our findings have significant implications for clinical diagnostics, One Health surveillance, and public health interventions.

## 1. Introduction

*Dirofilaria repens* (Spirurida, Onchocercidae) is a zoonotic vector-borne filarial parasite in Europe, transmitted by Culicidae mosquitoes, with dogs as the main reservoirs of infection [1]. Although human dirofilariosis is not a notifiable disease in many European countries, there is increasing evidence of its rising incidence across the continent [2]. The infection is expanding beyond traditionally endemic regions of southern Europe into previously unaffected northern areas [3,4]. *D. repens* typically localizes in the subcutaneous tissue, causing the formation of subcutaneous nodules or under the conjunctiva (ocular dirofilariosis), following transmission of the infecting larvae (L3) by mosquitoes [1]. Moreover, in specific cases, depending on the anatomical location and the affected tissue type, *D. repens* infection may encompass a broader range of clinical subcategories, including involvement of the oral cavity, lymph nodes, skeletal muscles, reproductive tract, female breast, and the lungs [5].

In recent years, a noticeable increase in reported human infections has been observed, with microfilariae detected either within excised adult worms or, more rarely, in peripheral blood. To date, 26 such cases have been reported [5,6], suggesting that in rare instances, *D. repens* may complete its life cycle in the human host. However, until now, no case has provided clear evidence of persistent microfilaremia. We report the first objectively confirmed case of long-lasting microfilaremia in a female patient from an endemic region in Croatia. This unique finding raises important questions about the potential for *D. repens* to establish patent infections in humans, the duration of microfilarial circulation, and the broader epidemiological implications.

## 2. Case Presentation

A 54-year-old woman from a rural area on the west coast of Istria (Croatia) was referred to the University Hospital for Infectious Diseases in June 2024 for evaluation of persistent peripheral eosinophilia. Initial laboratory findings revealed an absolute eosinophil count of 900/µL (14.2%), with subsequent tests confirming fluctuating eosinophilia over several months. The patient was otherwise afebrile, hemodynamically stable with no systemic symptoms. She reported an intermittent migratory rash on her limbs and trunk (Figure 1).

The clinical course began in January 2024 with the surgical excision of a subcutaneous nodule on her left upper arm that had persisted for several months. Through that time, the node had remained stable in size and nontender on palpation. The overlying skin appeared normal, with no signs of erythema, induration, or other pathological changes. Histopathological examination revealed a reactive lymph node with necrotic zones, eosinophilic infiltration, and parasitic structures. Although parasitological evaluation was recommended, it was not conducted.

The patient underwent an initial examination and management at the Infectious Diseases Department of another Clinical Hospital center in February 2024. An abdominal ultrasound revealed a 2.2 cm hepatic cyst in the left lobe without lymphadenopathy. The patient had been aware of the cyst for over a year without progression. Serological tests for *Toxocara*, *Trichinella*, *Brucella*, and *Bartonella henselae* were negative, except for borderline *Echinococcus* IgG titers and markers of past exposure to *T. gondii*. HIV serology was negative. In mid-March, an ophthalmologic examination showed no evidence of ocular involvement. A complete blood count (CBC), performed at the beginning of April, confirmed eosinophilia (17%, absolute count 900/µL). A chest X-ray showed a 14 mm retrocardial pulmonary nodule. In view of the persistent eosinophilia, further parasitological evaluation was indicated. Albendazole therapy (2 × 400 mg/day) was administered for 7 days. Filaria serology was positive (27 NTU), and repeated serology for Echinococcus was negative. Chest CT confirmed the hepatic cyst (now 2.6 cm) but showed no active pulmonary pathology.

Due to persistent eosinophilia and symptoms, the patient sought a second opinion in June 2024. In the University Hospital for Infectious Diseases Zagreb. CBC showed mild eosinophilia (7.8%, absolute count 400/µL). In August 2024, she reported fatigue, myalgia, and a migratory skin rash described as burning and crawling. CBC revealed persistent eosinophilia (16.7% absolute count 900/µL), with increased Filaria antibody titer (44 NTU). Three microfilariae were detected upon examination of multiple thick blood smears (Figure 2).

In late October, she again reported a transient erythematous rash over the site of previous nodal excision and a sensation of subcutaneous movement. She felt tired and had mild myalgia during the rash outbreak. Despite these symptoms, she remained afebrile with maintained appetite, with no respiratory symptoms or arthralgia, normal bowel and urinary habits, and stable body weight.

In October 2024, a transient erythematous rash appeared at the previous surgical site, together with a subjective sensation of subcutaneous movement, while no other clinical signs were observed. In December 2024, due to ongoing symptoms, she was hospitalized for further evaluation. The modified Knott’s test detected five microfilariae in 5 mL of blood in October and one microfilaria in 3 mL in December, corresponding to 1.0 and 0.33 mf/mL, respectively. The morphology was consistent with *Dirofilaria* spp. (Figure 2 and Figure 3).

Sequencing of the *COI* (Cytochrome c oxidase subunit I) gene confirmed the presence of *D. repens* from DNA extracted from blood and paraffin-embedded tissue [7,8]. In addition, *Wolbachia* sp. endosymbionts of *D. repens* were detected in the paraffin blocks [9] and microfilariae. Despite confirmed transient microfilaremia, the absence of active microfilariae in repeated smears and normalized eosinophil count (EO abs 400/µL) led to a decision not to initiate ivermectin treatment. Instead, Doxycycline therapy (2 × 100 mg/day for 14 days) was initiated in December.

During hospitalization, the patient remained clinically stable, with no fever, organ dysfunction, or new symptoms. She was discharged with instructions for regular follow-up, including monthly CBC and liver enzyme tests, serological follow-up for filarial markers, and a control ophthalmological examination. As of January 2025, she remained afebrile, with improved subjective well-being and no recurrence of cutaneous symptoms.

In February 2025, she presented with transient granuloma-like changes on the left upper eyelid, which resolved spontaneously. As of July 2025, at the last follow-up visit, she remains well, with an eosinophil count within the reference range (6.5%, 280/µL). Annual ultrasound monitoring of the liver cyst and repeat serology are planned. To facilitate tracking of diagnostic procedures, treatment, and clinical events, a timeline was constructed summarizing the patient’s course from initial presentation to follow-up (Figure 4).

## 3. Discussion

This case further supports the growing body of evidence that humans may not always represent dead-end hosts for *Dirofilaria repens*. Although most human infections manifest as subcutaneous or ocular nodules, a small but increasing number of reports indicate systemic involvement, including microfilaremia [5]. To date, approximately 12 cases of human microfilaremia due to *D. repens* have been reported worldwide, involving transient or single-timepoint detections. In contrast, our case demonstrates persistent microfilaremia confirmed by repeated microfilaremia over several months, making it the first documented case of long-term patent infection. This persistence, rather than transient circulation, distinguishes our observation and highlights the need to reconsider the biological potential of *D. repens* in human hosts. Although human dirofilariasis cases have been increasingly reported [10], in this particular case, suspicion of *D. repens* infection arose only six months after surgical removal of an adult worm. The diagnosis was finally confirmed by the detection of microfilariae in a thick blood smear and through *COI* gene sequencing from both the microfilariae (following a positive Knott’s test) and the paraffin-embedded tissue block. The delay in diagnosis may, in part, be attributed to the non-specific nature of the clinical signs, as the erythema, rash, and discomfort experienced by the patient could be attributed to a variety of other conditions. Nevertheless, in this case, these dermatological changes were likely associated with the presence of microfilariae. Similar skin manifestations have been described in dirofilariasis, although they are most often linked to the activity of adult worms [11]. The detection of *Wolbachia* endosymbionts provided additional confirmation of an active infection.

Remarkably, microfilaremia in this patient was documented over a four-month interval (August–December 2024), with repeated detection of up to three microfilariae on wet-mount peripheral blood smears and microfilarial loads of 1.0 and 0.33 mf/mL by the modified Knott’s test in October and December, respectively. The microfilaremia may have persisted for a longer period, given the surgical removal of an adult worm in February 2024; however, the absence of testing on earlier blood samples precludes definitive conclusions. Interestingly, microfilariae were initially detected using thick blood smears, whereas in the two subsequent examinations, they were identified only through Knott’s test but not by thick smear analysis. This discrepancy likely reflects a higher microfilarial load at the time of initial detection, with subsequent decline to levels detectable only through concentration technique. Persistent peripheral eosinophilia of up to 20% during this period may indicate a higher reproductive activity of the parasite in the human host than previously recognized.

In contrast to previous findings, where microfilariae disappeared rapidly following surgical removal of the adult parasite, this case demonstrates sustained microfilaremia for the first time in humans [6]. While the patient resided in an endemic area and had frequent mosquito exposure, her dog tested negative for microfilariae on two separate occasions, unlike a previously reported case in which the patient’s dogs had high microfilaremia (5000–7000 mf/mL) [6]. This finding highlights the possibility that human infections may occur independently of household reservoirs, underscoring the importance of broader One Health surveillance, including environmental and vector-based components.

From a diagnostic standpoint, this case emphasizes the importance of performing the Knott’s test on at least 5 mL of blood to enhance sensitivity. In endemic regions, *D. repens* should be considered in the differential diagnosis of unexplained eosinophilia. Longitudinal follow-up, including repeated Knott tests and eosinophil count monitoring, is recommended when clinical suspicion remains high.

In our case, ivermectin was considered but ultimately not used. In the absence of evidence for additional adult *D. repens* worms, and given the documented anti-wolbachial activity of doxycycline and thereby reducing microfilaremia, we opted for doxycycline therapy only [12,13]. This approach achieved the intended outcome, with microfilariae no longer detectable at follow-up. Had microfilariae persisted, we would have added ivermectin.

Overall, this case contributes important new evidence that under certain conditions, *D. repens* may complete its life cycle in humans. It reinforces the need for increased clinical awareness, molecular confirmation, and integrated vector surveillance to better understand and manage this emerging zoonotic infection. Human dirofilariasis represents a textbook example of a One Health disease: transmission to humans occurs via mosquito vectors, while dogs—typically asymptomatic carriers—serve as the primary reservoir and are often left untreated [5,6]. The rising number of human cases highlights the urgent need to include dirofilariasis on the list of notifiable diseases, enabling evidence-based planning and implementation of targeted interventions aimed at breaking the transmission cycle. This increase is most likely linked to the spread of invasive mosquito species, which are competent vectors for *D. repens* [14].

By contrast, with strictly anthroponotic filariases such as *Loa loa* and *Onchocerca volvulus* human *D. repens* infection rarely produces microfilaremia and, when present, it is of very low number and short duration, making onward transmission to humans unlikely. Within this context, our case is documented, persistent microfilaremia, which may challenge the prevailing view that humans serve exclusively as dead-end hosts and underscores the need to investigate host–parasite–vector conditions that could permit sustained microfilarial production in humans.

Recent surveillance highlights the expanding distribution of *Aedes albopictus* and *Aedes japonicus* in Central and Eastern Europe, species competent for *Dirofilaria* spp. transmission. The northward spread of both *Aedes* species parallels the documented increase in autochthonous human and canine *D. repens* cases in previously non-endemic areas [4]. This ecological overlap reinforces the potential for unexpected clinical presentations such as ours.

## 4. Conclusions

From a parasitological standpoint, this report provides novel evidence that *D. repens* can maintain reproductive activity in the human host for prolonged periods, resulting in sustained microfilaremia. This challenges the traditional view of humans as accidental dead-end hosts and suggests that, under favorable ecological and host-related conditions, the parasite’s life cycle may be completed in humans. Such findings raise important questions regarding host adaptation, vector competence, and the potential for changes in the epidemiology of *D. repens*. Given the growing environmental pressures, a rise in cases in which *D. repens* completes its life cycle in humans can be anticipated. Ultimately, this may lead to the parasite adapting to new host species—in this case, *Homo sapiens*.

## Figures and Tables

**Figure 1 microorganisms-13-02263-f001:**
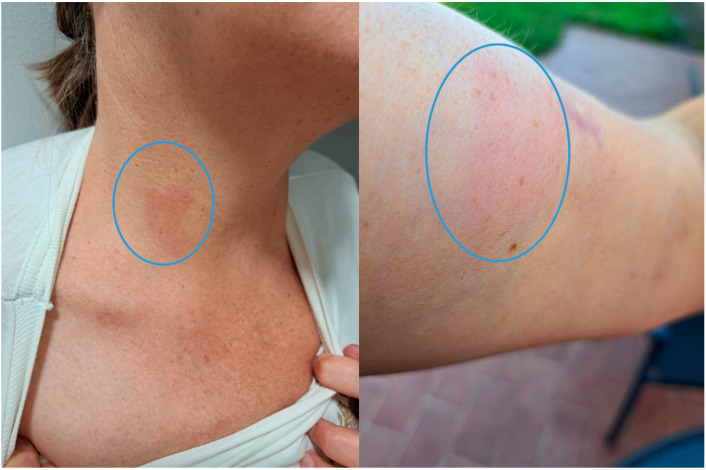
Erythema and rash on the neck (**left**) and arm (**right**), likely associated with the presence of *D. repens* microfilariae (circles indicates the lesion).

**Figure 2 microorganisms-13-02263-f002:**
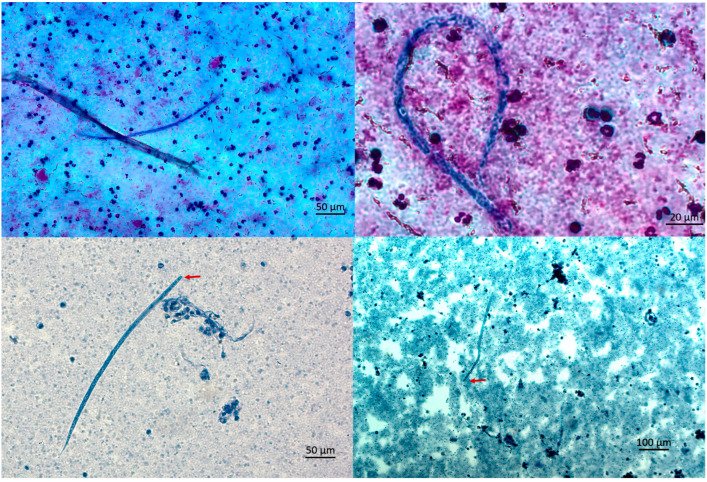
Microfilariae of *D. repens* observed in thick bloodsmear (**upper panel**) and after concentration by Knott’s test (**lower panel**). Arrows indicate anterior parts.

**Figure 3 microorganisms-13-02263-f003:**
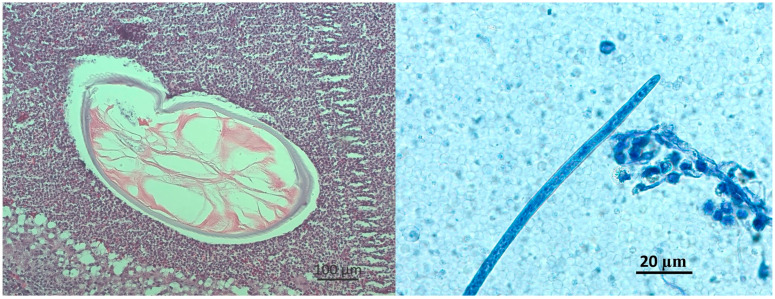
Histological section of a subcutaneous nodule showing a transverse section of an adult worm within host tissue (**left**). The parasite displays a thick multilayered cuticle and internal musculature. Anterior (cephalic) end of a *D. repens* microfilaria on Knotts test (**right**).

**Figure 4 microorganisms-13-02263-f004:**
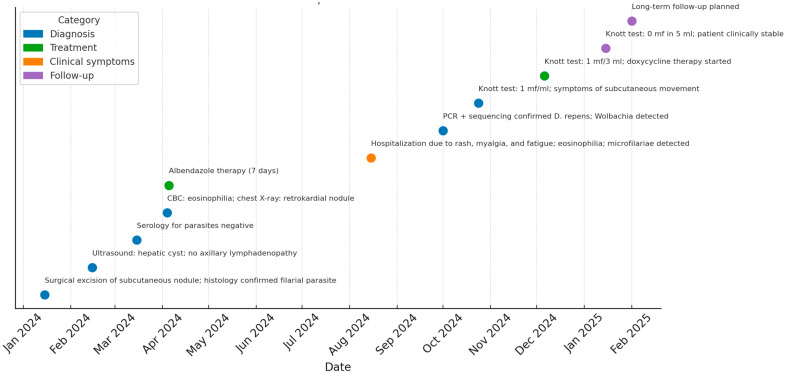
Timeline of Clinical Course. The timeline includes key dates for symptom onset, diagnostic testing (thick smear, Knott’s test, molecular confirmation), therapeutic interventions (albendazole, doxycycline), and laboratory follow-up (eosinophil counts, serology), providing a clear overview of the progression and management of this unique case.

## Data Availability

The original contributions presented in this study are included in the article. Further inquiries can be directed to the corresponding author.

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
