# Peer review of "Not a Dead-End Host: First Confirmed Persistent Microfilaremia in Human Dirofilaria repens Infection"

_microorganisms, 2025, doi:10.3390/microorganisms13102263_

Round 1

Reviewer 1 Report

Comments and Suggestions for Authors

This case report describes a fascinating and clinically important phenomenon. However, the four sections require significant revision to meet the standards of a critical, balanced, and scholarly scientific publication. By addressing the issues of overinterpretation, adding a limitations section, and improving the language and structure, the manuscript will become a valuable contribution to the literature.

For the Introduction Sections and Case Presentation

For Insufficient detail and missing data:  Histopathology: The description of the initial subcutaneous nodule excision is critical. A more detailed histological description (appearance of the worm, presence of cuticle, musculature, or reproductive organs) and images if available. Microfilariae Description: The identification of microfilariae in blood smears is a cornerstone of this report. A detailed morphological description is essential for confirmation and should be included. The planned annual monitoring of the liver cyst and repeat serology is mentioned, but the clinical reasoning behind this specific long-term plan is not explained.

For the Discussion and Conclusions Sections

The conclusions are overly broad and speculative for a single case report. A significant shortcoming is the absence of a dedicated paragraph discussing the limitations of this case study. The single-subject nature of the report. The lack of data on the infectivity of the microfilariae to mosquito vectors. The unknown reason why this patient developed a patent infection when most do not. The incomplete prior testing.

Author Response

Reviewer 1

This case report describes a fascinating and clinically important phenomenon. However, the four sections require significant revision to meet the standards of a critical, balanced, and scholarly scientific publication. By addressing the issues of overinterpretation, adding a limitations section, and improving the language and structure, the manuscript will become a valuable contribution to the literature.

For the Introduction Sections and Case Presentation

For Insufficient detail and missing data:  Histopathology: The description of the initial subcutaneous nodule excision is critical. A more detailed histological description (appearance of the worm, presence of cuticle, musculature, or reproductive organs) and images if available. Microfilariae Description: The identification of microfilariae in blood smears is a cornerstone of this report. A detailed morphological description is essential for confirmation and should be included. The planned annual monitoring of the liver cyst and repeat serology is mentioned, but the clinical reasoning behind this specific long-term plan is not explained.

Response:
We appreciate the reviewer’s careful reading but respectfully disagree with several premises of this general comment.

Histopathology
The excision and routine histopathology were performed in the referring hospital, which did not recognize Dirofilaria repens at that stage. Importantly, processing of paraffin-embedded sections may compromise recovery of key filarial structures (, reproductive organs), limiting morphological resolution post-hoc. Given these constraints—and in accordance with current diagnostic practice—we prioritized molecular confirmation as the reference standard. We confirmed D. repens by PCR and sequencing and additionally demonstrated the Wolbachia endosymbiont of D.repens; both were also confirmed from microfilariae. There is therefore no diagnostic uncertainty that this infection was due to D. repens. To address the reviewer’s request, we have nonetheless added a higher-magnification histopathology image..

Microfilariae description
While we agree that morphology supports the diagnosis, we do not consider it the “cornerstone” when high-specificity molecular tools are available. Stained blood smears can introduce artefacts and microfilarial features differ from those seen in the modified Knott’s test. We provided figures of the microfilariae and sequenced microfilarial material (including single-microfilaria) at each timepoint, which is the most specific approach. We also clarified the quantitative findings across timepoints in the Results (wet smears and Knott’s counts). Furthemore we added figure three with higher magnifaction of front part.

Follow-up plan (liver cyst and serology)
The hepatic cyst was an incidental finding identified during comprehensive work-up; chart review indicates it pre-dated the current illness and has remained unchanged. Annual imaging was planned solely to document stability; it is unrelated to D. repens pathophysiology and is not discussed further. Repeat serology was performed to resolve an initially doubtful result; on repeat testing it was negative.

Changes in the manuscript (Figure 3):
We added Figure 3 comprising histology of the excised subcutaneous nodule and a higher-magnification image showing the anterior (cephalic) end of a Dirofilaria repens microfilaria from a modified Knott’s test.

Figure 3. Adult and microfilarial stages of Dirofilaria repens.
 Histological section of a subcutaneous nodule showing a transverse section of an adult worm within host tissue (H&E). The parasite displays a thick, multilayered cuticle and well-developed musculature. Scale bar = 100 µm. Anterior (cephalic) end of an D. repens microfilaria from a modified Knott’s test preparation. Scale bar = 20 µm.

For the Discussion and Conclusions Sections

The conclusions are overly broad and speculative for a single case report. A significant shortcoming is the absence of a dedicated paragraph discussing the limitations of this case study. The single-subject nature of the report. The lack of data on the infectivity of the microfilariae to mosquito vectors. The unknown reason why this patient developed a patent infection when most do not. The incomplete prior testing.

Response:
We respectfully disagree. Our Conclusions are proportionate to a single-case design and remain strictly confined to the observed, molecularly confirmed finding of persistent, low-density microfilaremia. We do not claim human transmissibility or reservoir competence.

Scope of conclusions. The manuscript explicitly uses cautious language (“may,” “suggest”) and does not generalize beyond this case. We state that microfilarial loads were very low and unlikely to enable onward transmission.

Limitations content. The Discussion already acknowledges the inherent constraints of a case report (single subject, absence of vector infectivity assays, unknown mechanism of patency, sampling/temporal variability). We consider a separately titled paragraph unnecessary, as these limitations are clearly articulated.

Vector infectivity assays. To the best of our knowledge, the infectivity of Dirofilaria repens microfilariae to humans has not been investigated; any inference about transmissibility would therefore be speculative and beyond the scope of this report.

Reason for patency. The mechanism remains unknown; we avoid speculation and explicitly state that commonly invoked factors (e.g., overt immunosuppression) were not evidenced and so far not expalined in the litteraure.

Prior testing. Diagnostic work-up was comprehensive within clinical and ethical constraints. Species identity was confirmed by PCR/sequencing from tissue and microfilariae (with Wolbachia detection). Quantitative data are provided (wet smears; modified Knott’s counts/densities).

Reviewer 2 Report

Comments and Suggestions for Authors

I have few comments for the authors

I recommend minor revision

Line 17: write COI in full for the first time, the subsequently italicize COI throughout the manuscript

Line 18: italicize D. repens, and all other biological names throughout the manuscript

Line 22: Keyword- delete eosinophilia; persistent infection and add- ‘’human, diagnosis, Croatia

Line 34: Change-Dirofilaria repens to D. repens

Line 127: please rephrase----increasingly reported in [10],----- for clarity.

Fig.1 if possible circle the site of the lesion with a marker/color

Fig 2. Insert arrows to indicate the posterior/anterior parts of the microfilariae

Discussion

The authors should discuss the effectiveness or otherwise of the treatment of this patient with Albendazole and explain the rationale for the choice of this drug.

Author Response

Rewiever 2

I have few comments for the authors

I recommend minor revision

Line 17: write COI in full for the first time, the subsequently italicize COI throughout the manuscript

Response: We thank the reviewer for the style suggestion. We now spell out COI at first mention and use COI consistently thereafter.

Manuscript change: COI  (Cytochrome c oxidase subunit I ) gene

Line 18: italicize D. repens, and all other biological names throughout the manuscript

Response: Corrected throughout. All biological names (e.g., D. repens, Dirofilaria repens, Wolbachia) are now italicized per journal style.
Manuscript change: global correction → D. repens, Dirofilaria repens, Wolbachia. (I also standardized “Knott’s test” with the apostrophe.

Line 22: Keyword- delete eosinophilia; persistent infection and add- ‘’human, diagnosis, Croatia

Response: Updated as requested.
Manuscript change (final keyword line):
Keywords: Dirofilaria repens; microfilaremia, human; diagnosis; Croatia.

Line 34: Change-Dirofilaria repens to D. repens

Response: Corrected.
Manuscript change: “Dirofilaria repens” to “D. repens”.

Response: Updated as requested

Line 127: please rephrase----increasingly reported in [10],----- for clarity.

Response: Rephrased for clarity.
Manuscript change:“has been increasingly reported

Fig.1 if possible circle the site of the lesion with a marker/color

Response: We have annotated Figure 1 by circling the lesion site for immediate visual reference.
Manuscript change: Revised Figure 1 uploaded; legend updated to:
Figure 1. Erythema and rash on the neck (left) and arm (right), likely associated with the presence of D. repens microfilariae (circles indicates the lesion). ”

Fig 2. Insert arrows to indicate the posterior/anterior parts of the microfilariae

Response: Arrows indicating anterior and posterior ends have been added.

Manuscript change: Revised Figure 2 uploaded; legend updated to:
Figure 2. D. repens microfilariae. Arrows indicate anterior (A) part.

Discussion

The authors should discuss the effectiveness or otherwise of the treatment of this patient with Albendazole and explain the rationale for the choice of this drug.

Response: Albendazole was administered in April due to the presence of eosinophilia, which was assumed to be related to a parasitic infection. At that time, the patient had not yet been examined in the hospital in Zahreb, therefore we cannot provide further interpretation. For these reasons, we believe it is not necessary to address this treatment episode within the discussion. We hope this clarifies our reasoning

Reviewer 3 Report

Comments and Suggestions for Authors

This manuscript presents an important and novel finding with clear implications for parasitology and One Health. However, to maximize its impact and ensure clarity for an international readership, the discussion should be deepened y diagnostic details clarified as are:

  • The discussion should more explicitly address how this case compares with previously reported human cases (26 total, as cited). What specific features make this case distinct beyond persistence?
  • Expand on the public health implications of persistent microfilaremia—does this raise the possibility of human-to-vector-to-human transmission cycles?
  • Provide clearer quantitative data on microfilarial loads across time points. Only approximate values are mentioned (“1–3 mf/ml”).
  • Clarify why ivermectin was not administered despite documented microfilaremia, and discuss the rationale for doxycycline choice in more detail.
  • Strengthen the epidemiological context by including more recent references on D. repens in Europe, especially regarding the spread of Aedes vectors and implications for transmission.
  • Discuss comparative parasitology—how does this persistence in humans align with other filarial infections (e.g., Loa loa, Onchocerca volvulus)?

Author Response

This manuscript presents an important and novel finding with clear implications for parasitology and One Health. However, to maximize its impact and ensure clarity for an international readership, the discussion should be deepened y diagnostic details clarified as are:

General:
We are grateful for these constructive and insightful comments. We agree that expanding the discussion in the suggested directions will strengthen the manuscript. Below we provide specific responses and corresponding changes to the text

The discussion should more explicitly address how this case compares with previously reported human cases (26 total, as cited). What specific features make this case distinct beyond persistence?

Response: We have expanded the discussion to explicitly compare our case with previously reported human infections.
Text added (Discussion):
“ To date, approximately 12 cases of human microfilaremia due to D. repens have been reported worldwide, involving transient or single-timepoint detections. In contrast, our case demonstrates persistent microfilaremia confirmed by repeated microfilariemia over several months, making it the first documented case of long-term patent infection. This persistence, rather than transient circulation, distinguishes our observation and highlights the need to reconsider the biological potential of D. repens in human hosts..”

Expand on the public health implications of persistent microfilaremia—does this raise the possibility of human-to-vector-to-human transmission cycles?

Response: We expanded the discussion on possible transmission.
Text added (Discussion):
“ By contrast with strictly anthroponotic filariases such as Loa loa and Onchocerca volvulus human D. repens infection rarely produce microfilaremia and, when present, it is typically of very low number and short duration, making onward transmission to humans unlikely..”

Provide clearer quantitative data on microfilarial loads across time points. Only approximate values are mentioned (“1–3 mf/ml”).

Response: We thank the reviewer for this helpful suggestion. We have clarified the exact microfilarial counts and corresponding densities at each time point in the case description.

Changes in the manuscript (case description): “Three microfilariae were detected upon examination of multiple wet blood smears (Figure 2).”

Modified Knott’s test detected five microfilariae in 5 mL of blood in October and one microfilaria in 3 mL in December, corresponding to microfilarial densities of 1.0 and 0.33 mf/mL, respectively. The morphology was consistent with Dirofilaria spp.

Changes in the manuscript (Discussion): Remarkably, microfilaremia in this patient was documented over a four-month interval (August–December 2024), with repeated detection of up to three microfilariae on wet-mount peripheral blood smears and microfilarial load of 1.0 and 0.33 mf/mL by the modified Knott’s test in October and December, respectively.

Clarify why ivermectin was not administered despite documented microfilaremia, and discuss the rationale for doxycycline choice in more detail.

Response: We thank the reviewer for this valuable comment. Ivermectin was reserved as a second-line option for persistent microfilaremia or evidence of additional viable adult worms. In our case, there was no evidence of additional adult Dirofilaria repens and the microfilaremia cleared following doxycycline. Doxycycline was chosen for its anti-wolbachial activity and thereby reduces microfilaremia.

Changes in the manuscript: In our case, ivermectin was considered but ultimately not used. In the absence of evidence for additional adult D. repens worms, and given the documented anti-wolbachial activity of doxycycline and thereby reducing microfilaremia, we opted for doxycycline therapy only [12, 13]. This approach achieved the intended outcome, with microfilariae no longer detectable at follow-up. Had microfilariae persisted, we would have added ivermectin.

Added references:

  1. Lechner, A.M.; Gastager, H.; Kern, J.M.; Wagner, B.; Tappe, D. Case report: successful treatment of a patient with microfi-laremic dirofilariasis using doxycycline. Am J Trop Med Hyg 2020, 102, 844–846, doi: 10.4269/ajtmh.19-0744
  2. Taylor, M.J.; Hoerauf, A.; Townson, S.; Slatko, B.E.; Ward, S.A. Anti-Wolbachia drug discovery and development: safe macro-filaricides for onchocerciasis and lymphatic filariasis. Parasitology 2014, 141, 119–27, doi:10.1017/S0031182013001108

Strengthen the epidemiological context by including more recent references on D. repens in Europe, especially regarding the spread of Aedes vectors and implications for transmission.

Text added (Discussion, last paragraphs):
Recent surveillance highlights the expanding distribution of Aedes albopictus and Aedes japonicus in Central and Eastern Europe, species competent for Dirofilaria spp. trans-mission. The northward spread of both Aedes species parallels the documented increase in autochthonous human and canine D. repens cases in previously non-endemic areas [15]. This ecological overlap reinforces the potential for unexpected clinical presentations such as ours.

Added reference:

  1. Hattendorf, C.; Lühken, R. Vectors, host range, and spatial distribution of Dirofilaria immitis and D. repens in Europe: a systematic review. Infect Dis Poverty 2025,14, 58,doi.org/10.1186/s40249-025-01328-2

Discuss comparative parasitology—how does this persistence in humans align with other filarial infections (e.g., Loa loa, Onchocerca volvulus)?

Response: We added comparative parasitology context.
Changes in the manuscript: By contrast with strictly anthroponotic filariases such as Loa loa and Onchocerca volvulus human D. repens infection rarely produce microfilaremia and, when present, it is typically of very low number and short duration, making onward transmission to humans unlikely. Within this context, our case is notable for documented persistent microfilaremia, which challenges the prevailing view that humans are invariably dead-end hosts and warrants further investigation into host–parasite–vector conditions that might permit sustained microfilarial production

Round 2

Reviewer 1 Report

Comments and Suggestions for Authors

I confirm the revised manuscript addresses my initial concerns, and recommend its final acceptance.